# Hepatitis B Core Protein Is Post-Translationally Modified through K29-Linked Ubiquitination

**DOI:** 10.3390/cells9122547

**Published:** 2020-11-26

**Authors:** Hana Langerová, Barbora Lubyová, Aleš Zábranský, Martin Hubálek, Kristýna Glendová, Ludovic Aillot, Jan Hodek, Dmytro Strunin, Václav Janovec, Ivan Hirsch, Jan Weber

**Affiliations:** 1Institute of Organic Chemistry and Biochemistry of the Czech Academy of Sciences, IOCB Gilead Research Center, 16000 Prague, Czech Republic; hana.langerova@uochb.cas.cz (H.L.); barbora.lubyova@uochb.cas.cz (B.L.); ales.zabransky@uochb.cas.cz (A.Z.); martin.hubalek@uochb.cas.cz (M.H.); kristyna.glendova@uochb.cas.cz (K.G.); ludovic.aillot@uochb.cas.cz (L.A.); jan.hodek@uochb.cas.cz (J.H.); dmytro.strunin@uochb.cas.cz (D.S.); vaclav.janovec@uochb.cas.cz (V.J.); hirschi@natur.cuni.cz (I.H.); 2Department of Genetics and Microbiology, Faculty of Science, Charles University, BIOCEV, 25250 Vestec, Czech Republic; 3Institute of Molecular Genetics of the Czech Academy of Sciences, 14220 Prague, Czech Republic

**Keywords:** hepatitis B virus, HBc, post-translational modifications, ubiquitination, ubiquitin, E3 ubiquitin-protein ligase

## Abstract

Hepatitis B virus (HBV) core protein (HBc) plays many roles in the HBV life cycle, such as regulation of transcription, RNA encapsidation, reverse transcription, and viral release. To accomplish these functions, HBc interacts with many host proteins and undergoes different post-translational modifications (PTMs). One of the most common PTMs is ubiquitination, which was shown to change the function, stability, and intracellular localization of different viral proteins, but the role of HBc ubiquitination in the HBV life cycle remains unknown. Here, we found that HBc protein is post-translationally modified through K29-linked ubiquitination. We performed a series of co-immunoprecipitation experiments with wild-type HBc, lysine to arginine HBc mutants and wild-type ubiquitin, single lysine to arginine ubiquitin mutants, or single ubiquitin-accepting lysine constructs. We observed that HBc protein could be modified by ubiquitination in transfected as well as infected hepatoma cells. In addition, ubiquitination predominantly occurred on HBc lysine 7 and the preferred ubiquitin chain linkage was through ubiquitin-K29. Mass spectrometry (MS) analyses detected ubiquitin protein ligase E3 component N-recognin 5 (UBR5) as a potential E3 ubiquitin ligase involved in K29-linked ubiquitination. These findings emphasize that ubiquitination of HBc may play an important role in HBV life cycle.

## 1. Introduction

Hepatitis B virus (HBV) is a hepatotropic virus belonging to the *Hepadnaviridae* family [1]. Chronic HBV disease leads to the development of liver diseases, including cirrhosis and hepatocellular carcinoma. Nowadays, the World Health Organization has revealed that an estimated 325 million people worldwide are chronically infected with hepatitis B or C viruses (HBV and HCV). Despite intensive research, available treatments, which are based on the application of nucleotide analogues and pegylated interferon, suppress viral replication but are not curative [2].

HBV persists by establishing an episomal covalently closed circular double-stranded DNA (cccDNA) from relaxed circular DNA in the nucleus of infected cells. cccDNA serves as a template for viral transcription [3,4] and expresses at least six overlapping RNAs transcribed from four open reading frames (ORFs): S, C, P, and X. The S ORF encodes surface envelope proteins (S, M, and L), the C ORF encodes the precore protein (external core antigen, HBeAg) and HBc protein (HBc), the P ORF encodes viral polymerase, and the X ORF encodes regulatory X protein [3,5,6].

HBc is a 183- or 185-aa protein of a length that varies depending on the viral genotype [7]. It is composed of an N-terminal (NTD HBc) and a C-terminal domain (CTD HBc), which are connected by a flexible linker. The NTD HBc is responsible for capsid assembly and CTD plays a critical role in the specific packaging of the viral pgRNA [8,9]. It has been shown that HBc is modified by different types of post-translational modifications (PTMs) [10]. PTMs control the fragile cellular homeostasis and their deviation leads to the development of human disease disorders, such as neurodegeneration [11], cardiovascular diseases [12], and cancer [13]. Among others, PTMs involve the addition of polypeptides (e.g., ubiquitination and ubiquitin-like-protein conjugation (UBL-protein)) [14].

Ubiquitination is driven by the small (8.6 kDa) regulatory protein ubiquitin, which mediates the process via covalent attachment of its glycine residue to the lysine residue of the target protein. This process is reversible, versatile, and dynamic [15]. Ubiquitination involves three types of enzymes: E1 ubiquitin-activating enzymes, E2 ubiquitin-conjugating enzymes, and E3 ubiquitin-ligase enzymes [16,17,18,19,20]. Recently, non-canonical sites of ubiquitination have been described. Among them, serine, threonine, cysteine, and tyrosine amino acid residues are potential targets of ubiquitin attack [21,22,23,24,25,26]. Ubiquitin itself contains seven lysine residues (K6, K11, K27, K29, K33, K48, K63) and one methionine residue (M1), through which it can be attached to the substrate or to another ubiquitin molecule [15]. The ubiquitin linkage specificity determines whether the target protein is degraded in the proteasomal or lysosomal pathway, or serves a different function within the cells [15,27].

Little is known about ubiquitination and UBL modifications of HBc. Rost et al. suggested in 2006 that the ubiquitin-interacting adaptor γ2-adaptin interacts with the lysine residue 96 (K96) of HBc and that this interaction is crucial for HBV egress from hepatocytes [28]. The authors also described a partial interaction between the PPAY-motif of HBc and the E3 ubiquitin-ligase NEDD4 inducing HBV production [28]. Garcia et al. showed that K-to-R mutations of either K7 or K96 lysine residues have no influence on HBV replication or virion release [29]. Further it has been shown that E3 ubiquitin-ligase NIRF (Np95/ICBP90-like RING finger protein) interacts directly with HBc. This interaction leads to HBc proteasome-mediated degradation [30]. Additionally, silencing of NIRF causes an increase of the HBc level, leading to the release of mature HBc particles [30]. Based on mass spectrometry analysis, we have previously found that the amino acid residues K7, S44, S49, T67, and S157 of HBc could serve as a target for ubiquitin or other ubiquitin-like modifications [31]. It has been shown that HBc is post-translationally modified by ubiquitin at lysine residue 7 (K7) in transfected HepG2-hNTCP cells. Other identified serine and threonine residues could be involved in so-called non-canonical ubiquitination [31,32].

UBL proteins, such as SUMO (small ubiquitin-like modifiers), ISG15 (interferon-stimulated gene 15), or NEDD8 (neuronal precursor cell expressed, developmentally downregulated 8), could modify other proteins because they display a significant sequence similarity with ubiquitin [33]. However, the involvement of ubiquitination, and UBL modifications of HBc in cells remains unknown. The identification of HBc PTMs can help to clarify their functions in the HBV life cycle.

Here, we revealed that K29-linked ubiquitination is a predominant type of HBc ubiquitination in an HBc-transfected hepatoma Huh7 cell line and HBV-infected HepG2-hNTCP cells. Surprisingly, little is known about the assembly of K29-linked ubiquitin chains and the biological role of this modification is not completely understood. We propose that lysine residue at position 7 of HBc is important for its ubiquitination. It seems, that not only lysine-based ubiquitination, but also the non-canonical ubiquitination takes place in HBc PTMs. MS analyses identified several cellular E3 ubiquitin-protein ligases that may be potentially responsible for this modification, such as UBR5.

## 2. Materials and Methods

### 2.1. Cells

HEK293T (human embryonic kidney cell line, ATCC, Manassas, VA, USA) and Huh7 cells (differentiated hepatocyte-derived carcinoma cell line, Japanese Collection of Research Bioresources Cell Bank, Ibaraki, Osaka, Japan) were cultured in Dulbecco’s Modified Eagle Medium (DMEM) supplemented with 10% fetal bovine serum (FBS, VWR, Radnor, PA, USA) and an antibiotic mixture (penicillin/streptomycin (PenStrep), Sigma-Aldrich, St. Louis, MO, USA) at 37 °C in a 5% CO_2_ atmosphere.

HepG2-hNTCP, a human liver cancer cell line HepG2 stably transfected with the human HBV entry receptor (sodium taurocholate co-transporting polypeptide, hNTCP), was obtained from Dr. Stephan Urban (Heidelberg University Hospital, Heidelberg, Germany). The cells were cultured in DMEM supplemented with 10% FBS, the antibiotic mixture (PenStrep), and puromycin (0.05 mg/mL, Sigma-Aldrich, St. Louis, MO, USA) at 37 °C in a 5% CO_2_ atmosphere.

HepG2.2.15 cells (a human liver cancer cell line HepG2 that harbors two head-to-tail dimers of the HBV genome, serotype ayw, genotype D, GenBank accession: U95551.1) were obtained from Dr. David Durantel (Cancer Research Center of Lyon, Lyon, France). The cells were cultured in DMEM supplemented with 10% FBS, antibiotic mixture (PenStrep), and G418 (final concentration of 0.4 mg/mL; Sigma-Aldrich, St. Louis, MO, USA). All three cell lines were mycoplasma negative (tested at Generi Biotech, Hradec Kralove, Czech Republic).

### 2.2. Reagents and Antibodies

Anti-HA-tag (rabbit polyclonal, MilliporeSigma, Burlington, MA, USA), anti-FLAG-tag (mouse monoclonal, MilliporeSigma, Burlington, MA, USA), anti-HA-tag (mouse monoclonal, MilliporeSigma, Burlington, MA, USA), anti-HBc (rabbit polyclonal, DAKO, Carpinteria, CA, USA), and anti-HBc (rabbit monoclonal, Gilead Sciences, Inc., Foster City, CA, USA) were used as primary antibodies. As secondary antibodies conjugated with horseradish peroxidase (HRP), we used goat anti-mouse (MilliporeSigma, Burlington, MA, USA) and goat anti-rabbit (MilliporeSigma, Burlington, MA, USA). For visualization, we used a SuperSignal™ West Femto Maximum Sensitivity Substrate (ThermoFisher Scientific, Waltham, MA, USA) and LAS-4000 imager. As secondary antibodies for the LI-COR system, we used goat anti-rabbit IgG (H + L) IRDye 800CW (LI-COR Biosciences, Lincoln, NE, USA) and goat anti-mouse IgG (H + L) IRDye 680RD (LI-COR Biosciences, Lincoln, NE, USA). Western blots were visualized using an LI-COR Odyssey CLx system and the Image Studio Lite Software (LI-COR Biosciences, Lincoln, NE, USA).

### 2.3. Plasmids

The full-length HBc (1–185 aa, genotype A, subtype adw2) expression plasmids tagged with FLAG, or HA and FLAG-tagged K-to-R mutations were generated as described previously [31]. HBc without tag was generated by PCR amplification of HBc ORF (as a template, we used plasmid pHY92CMV obtained from Dr. Huiling Yang (Gilead Sciences, Inc., Foster City, CA, USA)) followed by subcloning into pcDNA3.1 (ThermoFisher Scientific, Waltham, MA, USA). Its mutants HBc-K7R, HBc-K96R, and HBc-K7/96R plasmids were generated by site-directed mutagenesis (QuikChange II XL Site-Directed Mutagenesis Kit, Agilent Technologies, Santa Clara, CA, USA) using PCR. As primers for the mutagenesis of HBc-K7R, we used K7R-F 5′-ATGGACATTGACCCGTATAGAGAATTTGGAGCTACTGTGG-3′; K7R-R 5′-CCACAGTAGCTCCAAATTCTCTATACGGGTCAATGTCCAT-3′. As primers for the mutagenesis of HBc-K96R, we used K96R-F 5′-CTAACATGGGTTTAAGGATCAGGCAACTATTGTGG-3′; K96R-R 5′-CCACAATAGTTGCCTGATCCTTAAACCCATGTTAG-3′. The plasmids were verified by DNA sequencing (GATC Biotech AG, Konstanz, Germany).

pRK5-HA-Ubiquitin-WT (Ub-WT, #17608), pRK5-HA-Ubiquitin-K6 (Ub-K6, #22900), pRK5-HA-Ubiquitin-K11 (Ub-K11, #22901), pRK5-HA-Ubiquitin-K27 (Ub-K27, #22902), pRK5-HA-Ubiquitin-K29 (Ub-K29, #22903), pRK5-HA-Ubiquitin-K33 (Ub-K33, #17607), pRK5-HA-Ubiquitin-K48 (Ub-K48, #17605), pRK5-HA-Ubiquitin-K63 (Ub-K63, #17606), pRK5-HA-Ubiquitin-K0 (Ub-K0, #17603), pRK5-HA-Ubiquitin-K6R (Ub-K6R, #121153), pRK5-HA-Ubiquitin-K11R (Ub-K11R, #121154), pRK5-HA-Ubiquitin-K27R (Ub-K27R, #121155), pRK5-HA-Ubiquitin-K29R (Ub-K29R, #17602), and pRK5-HA-Ubiquitin-K48R (Ub-K48R, #17604) were obtained from Addgene (Addgene, Watertown, MA, USA; Table 1).

pRK5-HA-tagged Ubiquitin-K33R (Ub-K33R) and pRK5-Ubiquitin-K63R (Ub-K63R) plasmids were generated by site-directed mutagenesis (QuikChange II XL Site-Directed Mutagenesis Kit, Agilent Technologies, Santa Clara, CA, USA) using PCR. As primers for the mutagenesis of Ub-K33R, we used F 5′-GGGATGCCTTCCCTGTCTTGGATCTTTGCCTTGACA-3′ and R 5′-TGTCAAGGCAAAGATCCAAGACAGGGAAGGCATCCC-3′. As primers for the mutagenesis of Ub-K63R, we used F 5′-AGGGTGGACTCTCTCTGGATGTTGTAGTCAGACAGG-3′ and R 5′-CCTGTCTGACTACAACATCCAGAGAGAGTCCACCCT-3′. The plasmids were verified by DNA sequencing (GATC Biotech AG, Konstanz, Germany).

The MYC-DDK-tagged SUMO-1 (RC200633, SUMO-1) and MYC-DDK-tagged ISG15 (RC2012353, ISG15) plasmids were purchased from OriGene (Rockville, MD, USA). HA-SUMO-2 (SUMO-2, #48967) and HA-SUMO-3 (SUMO-3, #17361) were obtained from Addgene (Addgene, Watertown, MA, USA; Table 1). The pcDNA3.1 plasmid was obtained from ThermoFisher Scientific (Waltham, MA, USA).

### 2.4. Preparation of HBV

The HepG2.2.15 cell line was used for viral HBV production and purification. The virions were purified by 4% PEG8000 precipitation and centrifugation from collected cell-free supernatants.

### 2.5. HBV Infection of HepG2-hNTCP Cells and Their Transient Transfection

Two days before infection, the cells were incubated in medium supplemented with 2.5% DMSO and 5% FBS. The cells were infected with HepG2.2.15-derived HBV in the presence of 4% PEG8000 overnight (2000 viral genome equivalents per cell). 16 h later, the cells were washed 3 times with PBS and maintained in DMEM supplemented with 2.5% DMSO and 5% FBS. Four days post-infection, the cells were transiently transfected with a ubiquitin variant using Lipofectamine™ 3000 transfection reagent (ThermoFisher Scientific, Waltham, MA, USA) according to the manufacturer’s recommendation. 48 h post-transfection, the cells were harvested under denaturing conditions.

Two variants of HBc, ayw and adw2, were used in this study. Adw2 differs from ayw in CTD HBc, where two amino acid residues are inserted at position 153–154.

### 2.6. Transient Transfection

Huh-7 cells were transfected with a control plasmid (pcDNA3.1), plasmids expressing HA-Ubiquitin variants, or HBc-FLAG variants with canonical DYKDDDDK using the transfection reagent GenJet™ (SignaGen Laboratories, Rockville, MD, USA). The transient transfection was performed according to the manufacturer’s recommendation. Briefly, for the Huh7 cells, we used the recommended optimal ratio of GenJet™ (µL):DNA (µg) 3:1 in serum-free DMEM with high glucose. The cells were co-transfected with the appropriate DNA constructs in the ratio of 1:1. The prepared reaction mixture GenJet™/DNA complex was added dropwise onto the medium of transfected cells. The medium was changed 5 h post-transfection with fresh complete serum/antibiotics containing DMEM medium. In the day of harvesting, part of the cells was treated with the proteasome inhibitor MG132 (MilliporeSigma, Burlington, MA, USA) at a final concentration of 50 µM for 5 h at 37 °C and 5% CO_2_.

### 2.7. Sample Preparation

At 48 h post-transfection, Huh7 cells were washed with phosphate-buffered saline (PBS) and lysed under denaturing conditions in lysis buffer (2% SDS, 150 mM NaCl, 10 mM Tris-HCl, pH 8.0) supplemented with 10 mM NEM (*N*-ethylmaleimide, all Sigma-Aldrich, St. Louis, MO, USA), protease inhibitors (Protease inhibitor cocktail, Sigma-Aldrich, St. Louis, MO, USA), and phosphatase inhibitors (Phosphatase-Inhibitor-Mix I, Serva Electrophoresis GmbH, Heidelberg, Germany) [37]. The cell lysates were boiled for 10 min and each sample was 10x diluted with the dilution buffer (10 mM Tris-HCl, 150 mM NaCl, 2 mM EDTA, 1% Triton X-100, pH 8.0) supplemented with 10 mM NEM, protease, and phosphatase inhibitors. The samples were incubated at 4 °C for 30 min with rotation followed by centrifugation at 20,000× *g* for 30 min at 4 °C. The precleared supernatants were used for co-immunoprecipitation, BCA protein concentration measurement (Pierce™ BCA Protein Assay Kit, ThermoFisher Scientific, Waltham, MA, USA), and sample preparation for SDS-PAGE in protein loading buffer (100 mM Tris-HCl, 4% SDS, 3% beta-mercaptoethanol, 0.2% bromophenol blue, 20% glycerol, pH 6.8).

### 2.8. Co-Immunoprecipitation

For the co-immunoprecipitation of protein samples, 1 mg of total protein of each sample was used. Magnetic beads equilibrated in dilution buffer (anti-FLAG, MilliporeSigma, Burlington, MA, USA; or Pierce™ anti-HA, ThermoFisher Scientific, Waltham, MA, USA) were added to the total protein (17.5 µL magnetic beads/1 mg of total protein). The mixture was incubated at 4 °C with overnight rotation followed by four washes using washing buffer (10 mM Tris-HCl, 1 M NaCl, 1 mM EDTA, 1% NP-40, pH 8.0). Co-immunoprecipitated proteins were analyzed by Western blot using 4–20% gradient SDS-PAGE (Bio-Rad Laboratories, Inc., Hercules, CA, USA).

In the case of HBV-infected samples, 2 mg of total protein of each sample were used for co-immunoprecipitation. 25 µL of Protein A magnetic beads (PureProteome Protein A Magnetic Bead System, MilliporeSigma, Burlington, MA, USA) equilibrated with anti-HBc antibody obtained from Gilead Sciences were used for one sample. The mixture was incubated at 4 °C overnight with rotation, and the next day, the samples were washed and prepared for SDS-PAGE.

### 2.9. Western Blot

Samples were prepared by boiling in protein loading buffer, resolved by polyacrylamide gel electrophoresis (SDS-PAGE), and transferred onto a PVDF membrane. Immunoblotting was performed using primary antibodies and secondary antibodies conjugated with fluorophores using the LI-COR detection Odyssey CLx system and Image Studio Software. When using secondary antibodies conjugated with horseradish peroxidase, the detection was performed using a chemiluminescent substrate SuperSignal™ West Femto Maximum Sensitivity Substrate (ThermoFisher Scientific, Waltham, MA, USA) and the LAS-4000 imager.

### 2.10. HBeAg Detection by an Enzyme-Linked Immunosorbent Assay

HepG2-hNTCP cell culture supernatants from three different replicates were collected 2 days post-transfection and centrifuged at 300× *g* for 10 min to remove cellular debris. The samples were transferred to clean tubes and stored at −80 °C. The titer of HBeAg (ng/mL) was determined using a commercial ELISA kit (Bioneovan, Beijing, China) according to the manufacturer’s recommendation.

### 2.11. Liquid Chromatography-Tandem Mass Spectrometry Analysis

HepG2-hNTCP cells grown to 90% confluence in three T225 flasks were transiently transfected with the HBc-HA plasmid (or empty vector pcDNA3.1) using LipofectamineTM 3000 transfection reagent (ThermoFisher Scientific, Waltham, MA, USA) according to the manufacturer’s instructions. Cells producing HA-tagged HBc were cultivated for 34 h, treated for 4 h with 10 µM MG132 in fresh media, and harvested in lysis buffer containing 50 mM Hepes, 100 mM NaCl, 10% glycerol, 0.5% Nonidet P40, pH 7.9 (Co-IP buffer). The cells were lysed in Co-IP buffer for 1 h at 4 °C. Cellular debris was removed by centrifugation for 10 min at 15,000× *g* at 4 °C. HBc-HA was immunoprecipitated using 120 μL of anti-HA magnetic beads at 4 °C. Immunoprecipitates were washed three times with Co-IP buffer and three times with Co-IP buffer containing no detergents. HA-tagged HBc and its interacting proteins were eluted by HA peptide (0.5 mg/mL) in 100 μL, and the supernatant was separated from the beads and digested with trypsin overnight at pH 8.5. Experiments in HEK293T were done in the same way except for using five 100-mm plates per sample, X-tremeGENE HP DNA reagent (Roche Diagnostics, Basel, Switzerland) for transfection, and cells were harvested 24 h post-transfection.

The resulting peptides were separated on an UltiMate 3000 RSLCnano system (Thermo Fisher Scientific) coupled to a Mass Spectrometer Orbitrap Fusion and Fusion Lumos (Thermo Fisher Scientific). The peptides were trapped and desalted with 2% acetonitrile in 0.1% formic acid at a flow rate of 5 μL/min on an Acclaim PepMap100 column (5 μm, 5 mm by 300-μm internal diameter (ID); Thermo Fisher Scientific). Eluted peptides were separated using an Acclaim PepMap100 analytical column (2 μm, 50-cm by 75-μm ID; ThermoFisher Scientific, Waltham, MA, USA). The 125-min elution gradient at a constant flow rate of 300 nL/min was set to 5% phase B (0.1% formic acid in 99.9% acetonitrile) and 95% phase A (0.1% formic acid) for the first 1 min. Then, the content of acetonitrile was increased gradually. The orbitrap mass range was set from *m*/*z* 350 to 2000, in the MS mode, and the instrument acquired fragmentation spectra for ions of *m*/*z* 100 to 2000. A Proteome Discoverer 2.4 (ThermoFisher Scientific, Waltham, MA, USA) was used for peptide and protein identification using Sequest and Amanda as search engines and databases of sequences of HA-tagged HBc, Swiss-Prot human proteins (downloaded on 15 February 2016), and common contaminants. The data were also searched with MaxQuant (version 1.6.3.4, Max-Planck-Institute of Biochemistry, Planegg, Germany) and the same set of protein databases in order to obtain peptide and protein intensities applied at the label-free quantification (LFQ) step. Perseus software (version 1.650, Max-Planck-Institute of Biochemistry, Planegg, Germany) was used to perform LFQ comparison of three biological replicates of HA-tagged HBc cells and three biological replicates of cells transfected with empty vector.

To identify E3 ubiquitin-protein ligases potentially interacting with ubiquitin variants, Huh7 cells were transiently transfected with an empty vector (pcDNA3.1), HA-tagged Ub-WT, Ub-K29, or Ub-K29R using GenJet™ transfection reagent according to the manufacturer’s recommendation. 72 h after transfection, the cells were harvested in RIPA buffer (50 mM Tris-HCl, 150 mM NaCl, 1.0% IGEPAL CA-630, 0.5% sodium deoxycholate, 0.1% SDS, pH 8.0) supplemented with protease and phosphatase inhibitors. The cells were lysed on ice for 20 min at 4 °C and harvested. The samples were centrifuged for 10 min at 15,000× *g* at 4 °C and precipitated with anti-HA magnetic beads (3 mg of total protein/50 µL magnetic beads for each reaction) overnight. On the next day, the samples were washed four times with 10 mM Tris-HCl at pH 8.0. The immunoprecipitated complexes were analyzed by MS as described above. The processing of samples varied by changing two parameters, i.e., using a 70-min elution gradient and the database of sequences Swiss-Prot human proteins for analysis.

The mass spectrometry proteomics data have been deposited to the ProteomeXchange Consortium via the PRIDE [39] partner repository with the dataset identifier PXD021712 and 10.6019/PXD021712.

### 2.12. Statistical Analysis

Data were analyzed with Image Studio Lite Ver 5.2 (LI-COR Biosciences, Lincoln, NE, USA) and GraphPad Prism 8 (GraphPad Software, San Diego, CA, USA). Quantitative variables were expressed as means ± standard deviation of the mean (SD). Ordinary one-way ANOVA (analysis of variance) followed by the Dunnett’s test for multiple comparisons were used due to the nature of the data. *p* values of * *p* ≤ 0.05 and ** *p* < 0.01 were considered significant.

## 3. Results

### 3.1. HBc and Its Post-Translational Modifiers

First, we examined whether the UBL proteins SUMO-1, SUMO-2, SUMO-3, and ISG15 could covalently bind to and modify HBc. In contrast to our previous study performed in HepG2-hNTCP [31], the Huh7 cells were used in all co-transfection experiments of HBc with post-translational modifiers and different ubiquitin variants. To this end, we co-expressed UBL proteins (i.e., MYC-DDK-tagged SUMO-1, HA-tagged SUMO-2, HA-tagged SUMO-3, and MYC-DDK-tagged ISG15) with tagged HBc (i.e., FLAG-tagged or HA-tagged, depending on the used UBL protein) in Huh7 hepatocellular carcinoma cells. The stringent conditions were used during cell lysis, co-immunoprecipitation, and washing in order to identify the direct linkage of UBL proteins with tagged HBc [40]. Although co-immunoprecipitation of precleared cell lysates using anti-FLAG or anti-HA magnetic beads resulted in specific HBc co-precipitation in all samples, we were not able to detect any covalent modification of HBc involving SUMO-1,2,3 or ISG15 (Figure 1). These results indicate that no covalent binding and modification between HBc and the UBL proteins occurred. The modification of HBc with NEDD8 in Huh7 cells could not be evaluated because the expression of NEDD8 was below the detection limit (data not shown).

Then, to examine whether ubiquitin could modify HBc, we co-expressed HBc-FLAG WT (HBc-WT) and HA-tagged Ub-WT in cells and harvested them under denaturing conditions. Our unpublished results showed that both C-terminally HA- or FLAG-tagged HBc constructs were functional and were able to form the capsids (Langerova et al., unpublished). Proteins were immunoprecipitated using anti-FLAG magnetic beads and analyzed by Western blots. Due to the reversible nature of ubiquitination, all reaction solutions were supplemented with the deubiquitinating enzyme inhibitor NEM to prevent protein deubiquitination. To analyze the potential ubiquitin modification of HBc in hepatoma Huh7 under conditions of proteasome inhibition, we treated one set of samples with 50 µM MG132 before cell harvesting. Afterwards, the cells were harvested under denaturing conditions and the level of HBc ubiquitination was analyzed by immunoprecipitation with anti-FLAG antibodies followed by Western blot analysis determined by anti-HA antibodies (Figure 2). The treatment of samples with the inhibitor MG132 generally led to an artificial increase of the cellular level of ubiquitinated proteins. We found that ubiquitination in Huh7 cells yielded mono-ubiquitination (Ub_1_), di-ubiquitination (Ub_2_), tri-ubiquitination (Ub_3_), and polyubiquitination (Ub_n_) of HBc. The treatment with MG132 did not affect the pattern of ubiquitin chain formation.

To identify whether the HBc lysine residue at position 7 (K7) or 96 (K96) (Figure 3a) was responsible for the HBc ubiquitination, we transiently co-transfected Huh7 cells with FLAG-tagged HBc-WT, or single-to-double K-to-R HBc-FLAG variants (HBc-K7R, HBc-K96R, HBc-K7/96R) together with HA-tagged Ub-WT. 48 h after transfection, HBc was precipitated from precleared cell lysates using anti-FLAG magnetic beads. Immunoprecipitated complexes were analyzed by Western blots using an LI-COR Odyssey CLx system and Image Studio Lite Software (Figure 3b). Using the Western blot data, we normalized the HBc mutant’s ubiquitination (Ub_1_, Ub_2_, Ub_3_, and Ub_n_) to the protein blot signal intensities of precipitated HBc-WT (Figure 3c). HBc-WT intensities of Ub_1_, Ub_2_, Ub_3_, or Ub_n_ were set to 100% for comparison to each HBc mutant. HBc-K7R substantially reduced Ub_1_, Ub_2_, and Ub_3_ and slightly reduced Ub_n_ compared to HBc-WT (Figure 3c). HBc-K96R and HBc-K7/96R led to the decrease of Ub_1_, Ub_2_, and Ub_3_ and to the increase of Ub_n_ compared to HBc-WT (Figure 3c). The increase of Ub_n_ could be due to a rise of nonspecific ubiquitination of non-lysine residues. Because of the two lysine residues present in canonical FLAG-tag, we performed the same experiments with HBc variants without tag (Appendix A). 48 h post-transfection, ubiquitin was precipitated from precleared cell lysates and immunoprecipitated complexes were analyzed by Western blot. Our results show that the tag-less HBc ubiquitination yielded in Ub_1_, Ub_2_, Ub_3_, and Ub_n_ in the same manner as in Figure 3. As we observed the decrease of HBc ubiquitination also in the K7R mutant, we assume that FLAG-tag and its two lysine residues have no effect on HBc ubiquitination.

To quantify the Ub_1_, Ub_2_, Ub_3_, and Ub_n_ of all HBc variants, we transiently co-transfected the Huh7 cells with HBc-FLAG variants and a HA-tagged Ub-WT expressing plasmid. 48 h after transfection, the cells were harvested under denaturing conditions followed by co-immunoprecipitation using anti-FLAG magnetic beads. Immunoprecipitated complexes were analyzed by Western blots using the LI-COR Odyssey CLx system and Image Studio Lite Software, and statistical analysis of the protein blot signal intensity was performed based on results from three independent experiments (Figure 4). The data were normalized to the expression level of the respective HBc variant and the total ubiquitination of each HBc variant was set to 100%. We found that, compared to the level of HBc-WT, the levels of Ub_2_ and Ub_3_ in HBc-K7R significantly decreased (* *p* ≤ 0.05) while the level of Ub_n_ significantly increased (* *p* ≤ 0.05) (Figure 4b). In HBc-K7/96R, we observed a significant decrease of the levels of Ub_1_, Ub_2_ (* *p* ≤ 0.05), and Ub_3_ (** *p* < 0.01), and a significant increase of the level of Ub_n_ (** *p* < 0.01) compared to HBc-WT. Additionally, the Ub_3_ ratio was significantly lower in HBc-K96R (* *p* ≤ 0.05) compared to HBc-WT. All significant decreases of Ub_1_, Ub_2_, and Ub_3_ quantity in HBc-K7/96R compared to HBc-WT led to a significant increase in Ub_n_, which could be due to nonspecific ubiquitination of HBc. Taken together, these results (Figure 3 and Figure 4) suggest that the lysine residue at position 7 is crucial for HBc ubiquitination.

### 3.2. The Study of HBc Mutants and Ubiquitin Mutants

To determine which lysine residue of ubiquitin is responsible for the HBc ubiquitination, we performed a large series of co-immunoprecipitation experiments. We used both single K (Figure 5a) and K-to-R ubiquitin mutants (Figure 5b). Single K ubiquitin mutants contained only a single intact lysine residue at position 6, 11, 27, 29, 33, 48, and 63 while all other lysine residues were mutated to arginine residues. We also used lysine-free ubiquitin (Ub K0). In K-to-R ubiquitin mutants, only one lysine residue was substituted by an arginine residue and all other lysine residues were left intact. Huh7 cells were co-transfected with the HA-tagged ubiquitin variants (Ub-WT, Ub-K6, Ub-K11, Ub-K27, Ub-K29, Ub-K33, Ub-K48, Ub-K63, Ub-K0, Ub-K6R, Ub-K11R, Ub-K27R, Ub-K29R, Ub-K33R, Ub-K48R, Ub-K63R) and FLAG-tagged HBc variants (HBc-WT, HBc-K7R, HBc-K96R, HBc-K7/96R). All proteins were expressed in cells (Appendix A). Then, 48 h after transfection, the cells were harvested and processed for co-immunoprecipitation. The immunoprecipitated complexes were analyzed by Western blot using secondary antibodies conjugated with fluorophores in an LI-COR Odyssey CLx system (Figure 5c,d).

For Ub-K29 and Ub-WT, we observed similar patterns of HBc ubiquitination (top right and top left panels in Figure 5c, respectively), and the ubiquitination of HBc-K7R was also reduced in Ub_1_, Ub_2_, Ub_3_, and Ub_n_. The substitution of lysine 29 by an arginine residue in Ub-K29R led to a complete loss of Ub_1_, Ub_2_, and Ub_3_ in HBc (top right panel in Figure 5d) compared to Ub-WT (top left panel in Figure 5d). Only a light smear of Ub_n_ HBc detected in Ub-K29R (top right panel in Figure 5d) can supposedly be caused by a nonspecific or non-canonical ubiquitination of HBc. The appearance of additional HBc bands migrating above the expected size (in red only, thus without ubiquitination), seen with several mutants (Ub-K29, -K33, -K48, and K-63), suggests that the HBc-HBc dimer or other HBc oligomers resist denaturation conditions.

In Ub-K0 (bottom right panel in Figure 5c), the light smear of ubiquitination was still visible and the mutations K7R and K7/96R led to a decrease of Ub_1_. The trend of decreasing Ub_1_ in the HBc K7R mutant was observed not only in Ub-WT and Ub-K0 but also in the Ub-K27 and Ub-K29 single-K ubiquitin mutants (top right panel in Figure 5c). In the HBc ubiquitination of Ub-K33 (bottom left panel in Figure 5c), we only observed light smears of all HBc variants, which suggested that the lysine residue K33 was not responsible for the HBc ubiquitination. On the contrary, when the K-to-R ubiquitin mutants Ub-K6R, Ub-K11R, and Ub-K33R (top left and the middle and bottom left panels in Figure 5d) were used, we still observed Ub_1_, Ub_2_, Ub_3_, and Ub_n_ in all HBc variants. Additionally, Ub_1_ was decreased in HBc-K7R and in HBc-K7/96R, where K48 and K63 (bottom middle and right panels in Figure 5d) lysine residues were mutated to arginine residues (Ub-K48R and Ub-K63R). These results suggested that HBc is predominantly ubiquitinated through K29-linked chains. These observations confirmed that the K7 of HBc takes an important place for this HBc modification.

### 3.3. Ubiquitination of Endogenous HBc

To investigate whether endogenous HBc undergoes ubiquitination in the context of the whole HBV genome and other viral proteins, HBV-infected HepG2-hNTCP cells were transiently transfected four days post-infection with an empty vector, i.e., HA-tagged Ub-WT, Ub-K29, or Ub-K29R. The transient transfection of ubiquitin variants into infected cells was assayed because of the detection limits of the endogenous ubiquitin. Overexpression of ubiquitin did not influence HBV production in infected cells as concluded from secretion of HBeAg in cell-free supernatant. The presence of HBeAg in media was analyzed from three biological replicates and we did not observe any significant changes in its amount in ubiquitin-transfected compared to no ubiquitin-transfected HBV-infected cells (Appendix A).

The cells were harvested under denaturing conditions 48 h after transfection. Precleared cell lysates were immunoprecipitated with anti-HBc equilibrated Protein A magnetic beads and immunoprecipitated complexes were analyzed by Western blots (Figure 6).

We were able to detect the protein expression of immunoprecipitated endogenous HBc (IP) and input exogenous ubiquitin variants. Replacing Ub-WT with Ub-K29R led to a reduction of HBc polyubiquitination especially in the field of ubiquitin detection between 80 and 190 kDa. On the contrary, the polyubiquitination of HBc increased using Ub-K29 instead of Ub-WT. In agreement with a previous experiment (Section 3.2), these results support the hypothesis that K29-linked ubiquitination is predominant in HBc ubiquitination.

### 3.4. MS Analysis of Ubiquitin-Protein Ligases Potentially Involved in HBc Ubiquitination

To identify the E3 ubiquitin-ligases potentially involved in the K29-linked ubiquitination, we transiently transfected the hepatoma Huh7 cells with either an empty vector, or HA-tagged Ub-WT, Ub-K29, or Ub-K29R. We then co-immunoprecipitated them with anti-HA magnetic beads and determined the E3 ubiquitin-ligases of each tested variant using MS analysis (Appendix A). We identified 26 different E3 ubiquitin-protein ligases, among them those that have been previously described in the formation of K29-linked ubiquitination, i.e., E3 ubiquitin-protein ligase UBR5 (UBR5) and E3 ubiquitin-protein ligase Itchy homolog (ITCH) [41,42].

To explore the HBc relationship with the ubiquitination pathway at the proteome level, we co-immunoprecipitated HBc-HA and associated proteins from transfected HepG2-hNTCP cells treated with the proteasome inhibitor MG132 and employed shotgun LC-MS/MS analysis for subsequent protein identification. We compared the protein level based on the number of specific peptides and peptide spectra (in the form of the peptide-spectra matches) of HBc pull downs and control samples in the LC-MS/MS spectra of three biological replicates (Table 2). Due to the low stringency of the Co-IP buffer used to preserve even weak and transient protein interactions, we detected a total of 28 E3 ubiquitin-protein ligases present in the HBc-HA immunoprecipitated complexes. Among ligases highly abundant exclusively in HBc samples, as shown by statistical analysis using a volcano plot (Appendix A), we identified E3 ubiquitin-protein ligase UBR5, known to mediate K29-linked ubiquitination (Appendix A). We also performed similar Co-IP analysis of HBc-HA in HEK293T cells, in biological duplicates. The experiment in HEK293T cells resulted in a significantly higher background level since most of the identified 56 E3 ubiquitin-protein ligases were detected also in negative (no HBc) control samples (Appendix A). However, there is still a clear difference in protein levels between HBc-HA and control samples for some of the E3 ubiquitin-protein ligases as seen in Appendix A.

## 4. Discussion

Our data suggest that the biological function of ubiquitin K29 is directly linked to the ubiquitination of HBc. These results were supported by a subsequent investigation of the endogenous HBc ubiquitination in HBV-infected cells in the context of the whole HBV genome and the expression of all viral proteins. We were still able to observe a low signal of HBc polyubiquitination using Ub-K29R. Therefore, we conclude that ubiquitin chains involving other lysine residues could also assemble on the HBc protein, although K29-linked ubiquitination is a predominant type of HBc modification in HBV-infected cells.

We found no evidence that the UBL proteins SUMO 1–3 or ISG15 modify HBc. For the detection of these modifications, we used a protocol recommended for UBL modification [40]. It is possible, however, that the methods we used were not sufficiently sensitive to detect HBc-UBL–protein interaction. In the future, the influence of UBL proteins on the whole HBV virus should be investigated.

Using an immunoprecipitation assay followed by Western blot analysis on gradient gels, we could clearly differentiate between Ub_1_, Ub_2_, Ub_3_, and Ub_n_ species of HBc in hepatoma Huh7 cells. The intensity of HBc ubiquitination increased in the presence of a proteasome inhibitor MG132, but the ubiquitination pattern remained unchanged. HBc contains only two lysine residues K7 and K96, which are conserved across all HBV strains. The role of these two lysine residues was addressed in several studies; however, little is known about the function of K7 residue. Ponsel et al. mutated 52 amino residues within the N-terminal domain of the HBc based on the crystal structure of recombinant capsids [43]. K96A mutation blocked HBV release but had no effect on the formation of cytoplasmic nucleocapsids. However, the effect of K7A HBc mutation was not studied in this work. Further, Garcia et al. used K7R, K96R, K96A, and K7/96R mutants to explain the role of lysine residues in HBV replication and release [29]. Virion release of HBc-WT, -K7R, -K96R, and -K7/96R were comparable and the K96A mutant was less efficient compared to WT. Mutants K7R, K96R, and K7/96R of HBc as well as K96A did not disrupt the formation of cytoplasmic HBV capsids compared to WT. Mutant K7R did not change the nucleo-cytoplasmic distribution compared to WT; however, mutants K96R and K7/96R were accumulated in the nucleus [29]. We showed that an HBc K-to-R mutation at position 7 (K7R) led to the decrease in HBc Ub_1_, Ub_2_, Ub_3_, and Ub_n_. A K-to-R mutation of HBc at position 96 (K96R) affected the level of ubiquitination only marginally. The double HBc mutation K7/96R caused the decrease of Ub_1_, Ub_2_, and Ub_3_ and the increase of nonspecific Ub_n_. These results agree with Garcia et al., in that more conserved K-to-R mutation of K96 residue had no influence on virus replication or release [29].

We also concluded that in HBc-K7R, Ub_2_ and Ub_3_ were significantly decreased and Ub_n_ significantly increased compared to HBc-WT. These results indicate that the lysine residue at position 7 plays an important role in the process of HBc ubiquitination. The observed decrease in Ub_1_, Ub_2_, and Ub_3_ and the increase in Ub_n_ in the K7/96R double mutant could be due to the non-specific or non-canonical ubiquitination of HBc. Non-canonical ubiquitination on non-lysine residues (e.g., serine, threonine, or cysteine residues) might reflect the ability of the cell to ubiquitinate proteins that are lacking, or not exposing lysine residues. In an experiment, which involved all investigated ubiquitin variants (WT, single K only, K-to-R and lysine-free ubiquitin mutants) and HBc variants (HBc-WT, HBc- K7R, HBc-K96R, and HBc-K7/96R), we identified specific ubiquitin chains conjugated to HBc in transfected cells.

Our research was focused on the identification of E3 ubiquitin-ligases potentially involved in HBc ubiquitination, which could clarify the importance of HBc ubiquitination in cells. To date, 2 E1 activating-enzymes, about 40 E2 conjugated-enzymes, and over 600 E3 ubiquitin-ligases have been identified [44]. E3 ubiquitin-ligases play a key role in this cascade transferring the ubiquitin to specific substrate proteins. Using MS, we identified the E3 ubiquitin-ligases potentially interacting with HBc. We found that none of the previously suggested E3 ubiquitin ligases (i.e., ITCH [41], UBR5 [42], UBE3C (ubiquitin-protein ligase E3C) [45], and Hul5 (probable E3 ubiquitin-protein ligase HUL5) [46]) assembled on K29-linked chains. Our MS data also revealed UBR5, from HEPG2-hNTCP and HEK293T cells, and ITCH, only from HEK293T cells, E3 ubiquitin-protein ligases in cell lysates. There was a significant difference in the protein levels for K29-linked E3 ubiquitin-protein ligase UBR5. UBR5 catalyzes the K11-, K29-, and K48/K63-linked ubiquitination [42,47], which is in agreement with the prediction that ubiquitin dimers involving K6, K11, K29, K33, and K48 could be formed between two ubiquitin moieties [48,49,50,51]. ITCH catalyzes K29-, K48-, and K63-linked ubiquitination [41,47,52,53,54]. Moreover, ITCH is closely related to other identified ligases WWP1 and WWP2, with which it shares several substrates [55,56,57]. It was also described that ligases UBR5/HUWE1/UBR4 co-operate with ITCH to assemble K48/K63 branched chains [47]. We also identified some E3 ubiquitin-protein ligases in MS analyses from HEK293T and HEPG2-hNTCP cells, whose role in the HBc life cycle has not yet been described. For example, MYCBP2 could be involved in non-canonical ubiquitination [58]. MYCBP2 can act in concert with another identified ubiquitin ligase HUWE1 [59]. UBR5 together with TRIP12 regulates under physiological conditions chromatin ubiquitination [60]. Evaluation of the potential involvement of these E3 ubiquitin-protein ligases in HBc ubiquitination requires further study because its detection appears to be influenced by the HBc bait protein. In the context of other viruses, it has been recently published that the replication protein PB2 of influenza A viruses (IAVs) is non-proteolytically ubiquitinated through K29-linked chains [61]. This ubiquitination is driven by two multicomponent RING-E3 ubiquitin-ligases based on cullin 4 (CRL4) and it contributes to IAV infection and viral production [61].

To date, only two studies describing the involvement of NIRF and NEDD4 E3 ubiquitin-ligases in HBc PTMs have been published. NIRF interacting with HBc promotes its degradation by the ubiquitin proteasome pathway [30]. Contrarily, NEDD4 interacts with HBc via late domain (L-domain) PPAY (the part of two nearby L-domain sequences: PPAYRPPNAP) [28,29]. The motifs of L-domains are generally PPXY (where X is any amino acid residue), and P(T/S)AP. Strack et al. stated that virus release is enhanced by the ability of these L-domains to recruit ubiquitin ligase activity [62]. We assume that HBc ubiquitination predominantly occurs on the K7 amino acid residue, which lies within the tyrosine-based motif YXXΦ (where Y is tyrosine, X is any amino acid residue, and Φ represents a bulky hydrophobic residue). Thus, we assume that K7-linked ubiquitination does not have a role in capsid formation and that the motif _6_YKEF_9_, including the K7 residue, has an important role in HBc trafficking and virus release. It would be premature to speculate on this subject, because further work is needed to confirm our hypothesis. In general, tyrosine-based motifs are well characterized in many viruses and are so-called L-like domains. This domain may interact with cellular proteins to facilitate virus budding and release from cells [63,64,65,66,67,68]. Moreover, the study of Chou et al. described the requirement of endosomal sorting complexes required for transport (ESCRT) components for HBV release [69]. Further research is needed to determine the role of this domain in HBc function and its role in the life cycle of HBV.

In conclusion, we showed that the HBc protein could be modified by ubiquitination in transfected as well as in infected hepatoma cells. Ubiquitination occurred predominantly on HBc lysine 7 and the preferred ubiquitin chain linkage was through Ub-K29. Nevertheless, we do not exclude the involvement of other ubiquitin K residues in this process or the possibility of mixed ubiquitin chain formation. Mass spectrometry analyses detected potential E3 ubiquitin-ligases involved in the K29-linked ubiquitination, such as UBR5. Its role in the HBc ubiquitination and HBV life cycle requires further evaluation.

## Figures and Tables

**Figure 1 cells-09-02547-f001:**
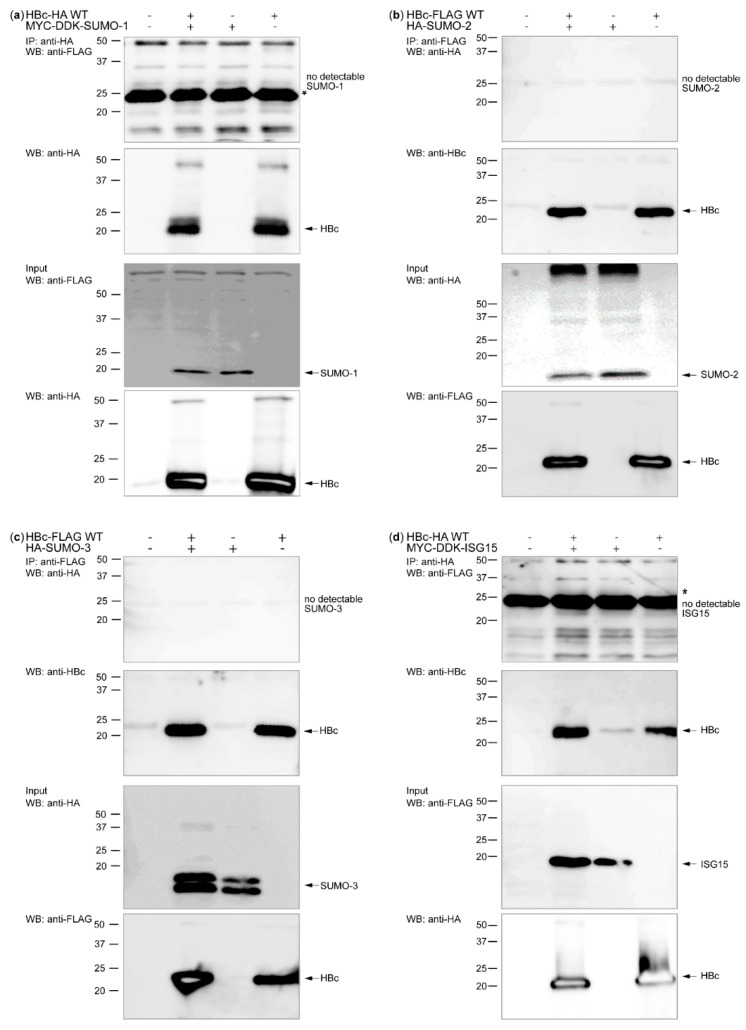
HBc is not post-translationally modified by UBL proteins SUMO 1–3 and ISG15. Western blots of immunoprecipitated cell lysates with anti-FLAG or anti-HA magnetic beads after transient co-transfection of Huh7 cells with HBc and the UBL proteins SUMO-1 (**a**), SUMO-2 (**b**), SUMO-3 (**c**), and ISG15 (**d**). The cells were harvested 48 h post-transfection and precleared cell lysates (1 mg of proteins, WB: Input 10 µg of total protein/lane) were immunoprecipitated with anti-FLAG, or anti-HA magnetic beads. HBc was detected using rabbit polyclonal and tags antibodies. The position of IgG light chains is indicated by *.

**Figure 2 cells-09-02547-f002:**
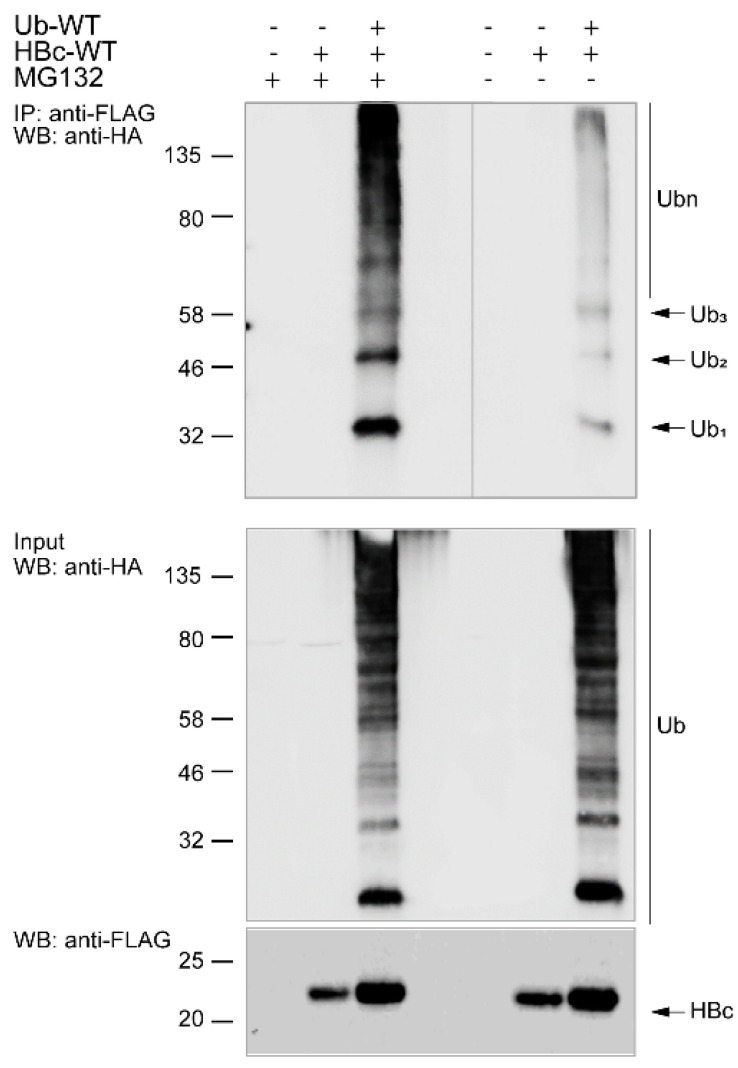
HBc is modified by ubiquitin in Huh7 cells. Western blots of cell lysates immunoprecipitated with anti-FLAG magnetic beads after co-transfection of Huh7 cells with FLAG-tagged HBc-WT and HA-tagged Ub-WT. The cells were treated 48 h post-transfection with 50 µM MG132 for 5 h. After the treatment, all cells were harvested and precleared cell lysates (1 mg of proteins, WB: Input 10 µg of total protein/lane) were immunoprecipitated with anti-FLAG magnetic beads. For HBc detection, the antibody against FLAG-tag was used.

**Figure 3 cells-09-02547-f003:**
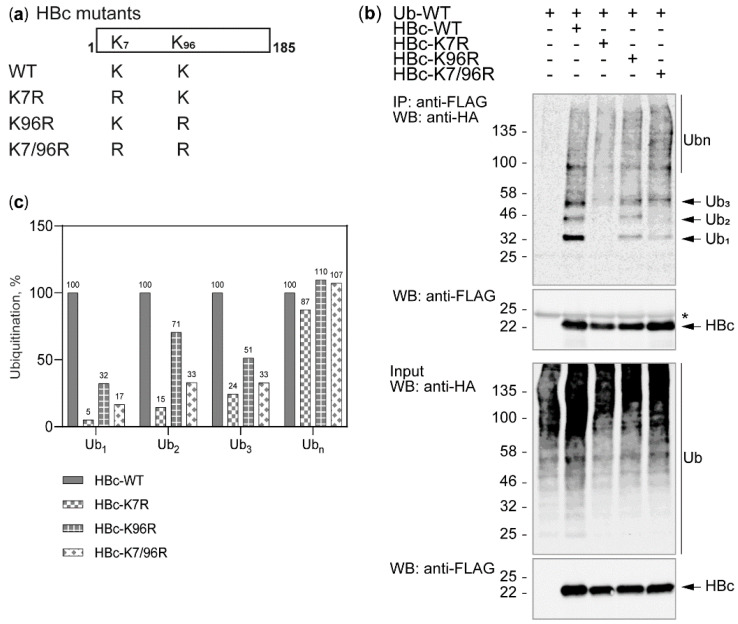
HBc-WT and its K-to-R mutants are ubiquitinated in different manners. (**a**) Graphical representation of single and double K-to-R HBc mutants; (**b**) Western blots of co-immunoprecipitated Huh7 cells through HBc using anti-FLAG magnetic beads after transient co-transfection with FLAG-tagged HBc-WT or single-to-double K-to-R HBc variants (HBc-K7R, HBc-K96R, HBc-K7/96R) and HA-tagged Ub-WT. The cells were harvested 48 h post-transfection and co-immunoprecipitation was performed through HBc using anti-FLAG magnetic beads. All proteins were expressed within the cells (WB: Input 10 µg of total protein/lane). Light chain of IgG is marked with *; (**c**) Relative ubiquitination of Ub_1_, Ub_2_, Ub_3_, and Ub_n_ of each HBc variant normalized to the level of their input and the HBc-WT input of the Western blot shown in (**b**).

**Figure 4 cells-09-02547-f004:**
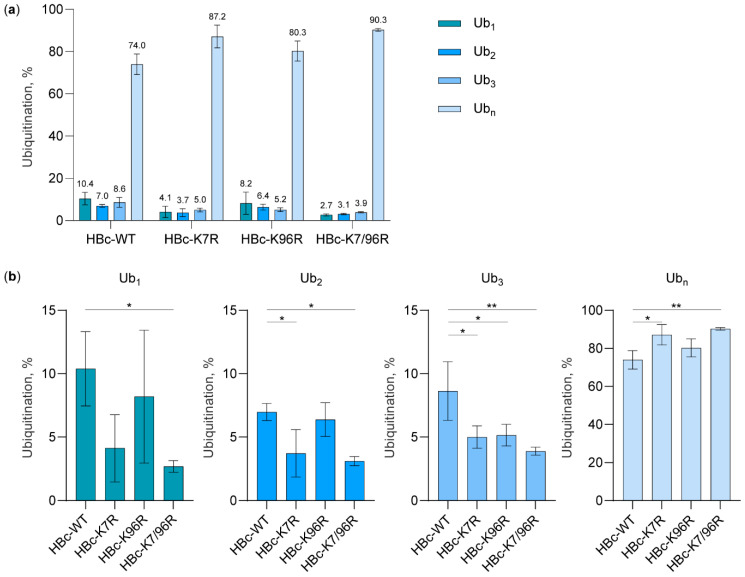
Lysine residue at position 7 is important for HBc ubiquitination. (**a**) Ratio of Ub_1_, Ub_2_, Ub_3_, and Ub_n_ in each HBc variant (HBc-WT, HBc-K7R, HBc-K96R, HBc-K7/96R); (**b**) Prevalence of each ubiquitination for different HBc variants (HBc-WT, HBc-K7R, HBc-K96R, HBc-K7/96R). The results are based on three independent experiments and normalized to the expression level of the respective HBc variant. The total ubiquitination of each HBc variant was set to 100%. Data is shown as mean ± standard deviation. * *p* ≤ 0.05; ** *p* < 0.01 obtained by ordinary one-way ANOVA followed by Dunnett’s test for multiple comparisons.

**Figure 5 cells-09-02547-f005:**
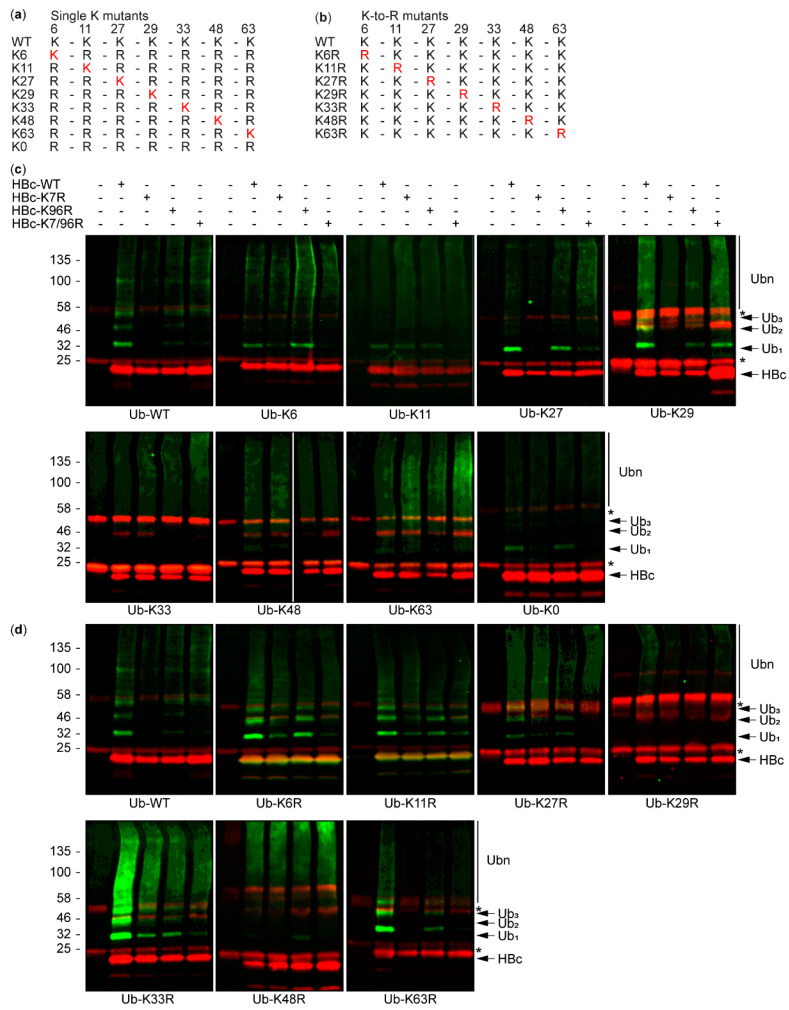
HBc variants are ubiquitinated at different levels using different ubiquitin variants. (**a**) Single K mutants and (**b**) K-to-R mutants used as ubiquitin variants; (**c**,**d**) Western blots of immunoprecipitated complexes of Huh 7 cells transiently co-transfected with one HA-tagged single K (**c**) or K-to-R ubiquitin mutant (**d**) and one FLAG-tagged HBc variant. For ubiquitin detection, the rabbit antibody against the HA-tag was used (IRDye 800CW, green), while for HBc detection, the mouse antibody against the FLAG-tag was used (IRDye 680RD, red). Original figure of Ub-K48 Western blot is included as Appendix A. Heavy and light chains of IgG are marked with *.

**Figure 6 cells-09-02547-f006:**
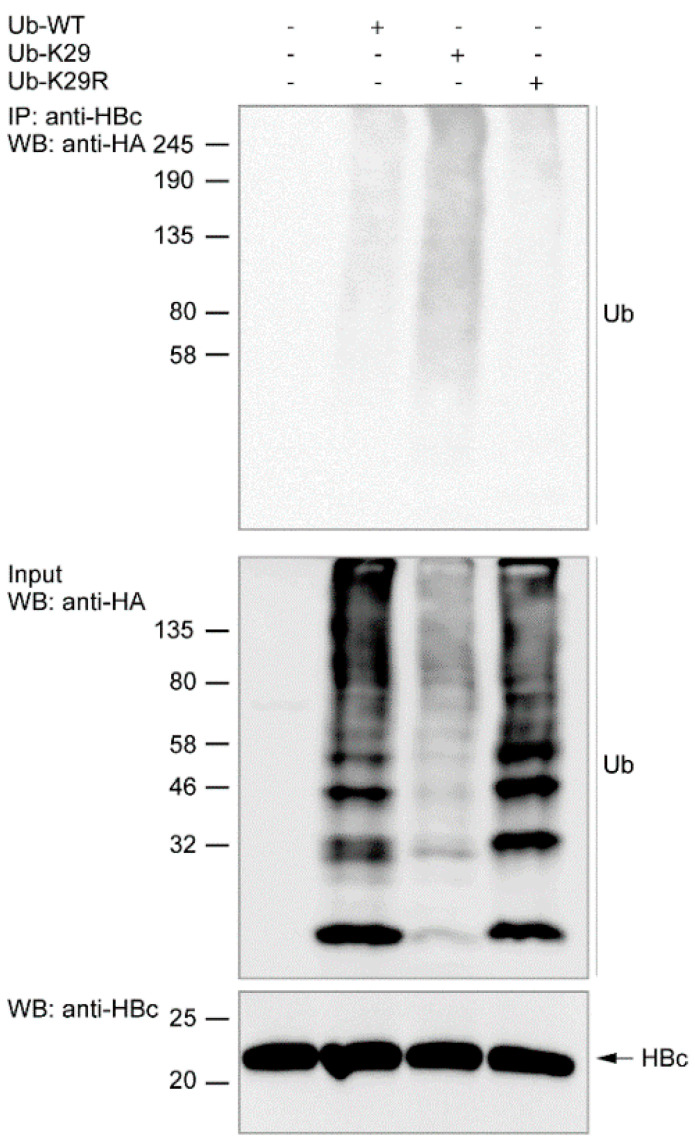
Endogenous HBc is polyubiquitinated within the HepG2-hNTCP cells. Western blots of co-immunoprecipitated complexes of HBV-infected HepG2-hNTCP cells transiently transfected with an empty vector, or HA-tagged Ub-WT, Ub-K29, or Ub-K29R with endogenous HBc and exogenous ubiquitin variants expressed in the cells (WB: Input of 10 µg of total protein/lane). Co-immunoprecipitation reactions were performed using Protein A magnetic beads equilibrated with anti-HBc antibody (from Gilead Sciences).

**Table 1 cells-09-02547-t001:** The list of plasmids obtained from Addgene.

Addgene Plasmid	Citation	Reference
pRK5-HA-Ubiquitin-WT	a gift from Ted Dawson (Addgene plasmid #17608; http://n2t.net/addgene:17608; RRID:Addgene_17608)	[34]
pRK5-HA-Ubiquitin-K6	a gift from Sandra Weller (Addgene plasmid #22900; http://n2t.net/addgene:22900; RRID:Addgene_22900)	[35]
pRK5-HA-Ubiquitin-K11	a gift from Sandra Weller (Addgene plasmid #22901; http://n2t.net/addgene:22901; RRID:Addgene_22901)	[35]
pRK5-HA-Ubiquitin-K27	a gift from Sandra Weller (Addgene plasmid #22902; http://n2t.net/addgene:22902; RRID:Addgene_22902)	[35]
pRK5-HA-Ubiquitin-K29	a gift from Sandra Weller (Addgene plasmid #22903; http://n2t.net/addgene:22903; RRID:Addgene_22903)	[35]
pRK5-HA-Ubiquitin-K33	a gift from Ted Dawson (Addgene plasmid #17607; http://n2t.net/addgene:17607; RRID:Addgene_17607)	[34]
pRK5-HA-Ubiquitin-K48	a gift from Ted Dawson (Addgene plasmid #17605; http://n2t.net/addgene:17605; RRID:Addgene_17605)	[34]
pRK5-HA-Ubiquitin-K63	a gift from Ted Dawson (Addgene plasmid #17606; http://n2t.net/addgene:17606; RRID:Addgene_17606)	[34]
pRK5-HA-Ubiquitin-K0	a gift from Ted Dawson (Addgene plasmid #17603; http://n2t.net/addgene:17603; RRID:Addgene_17603)	[34]
pRK5-HA-Ubiquitin-K6R	a gift from Josef Kittler (Addgene plasmid #121153; http://n2t.net/addgene:121153; RRID:Addgene_121153)	[36]
pRK5-HA-Ubiquitin-K11R	a gift from Josef Kittler (Addgene plasmid #121154; http://n2t.net/addgene:121154; RRID:Addgene_121154)	[36]
pRK5-HA-Ubiquitin-K27R	a gift from Josef Kittler (Addgene plasmid #121155; http://n2t.net/addgene:121155; RRID:Addgene_121155)	[36]
pRK5-HA-Ubiquitin-K29R	a gift from Ted Dawson (Addgene plasmid #17602; http://n2t.net/addgene:17602; RRID:Addgene_17602)	[34]
pRK5-HA-Ubiquitin-K48R	a gift from Ted Dawson (Addgene plasmid #17604; http://n2t.net/addgene:17604; RRID:Addgene_17604)	[34]
pcDNA3 HA-SUMO2 WT	a gift from Guy Salvesen (Addgene plasmid #48967; http://n2t.net/addgene:48967; RRID:Addgene_48967)	[37]
pcDNA3/HA-SUMO3 (Sentrin 2)	a gift from Edward Yeh (Addgene plasmid #17361; http://n2t.net/addgene:17361; RRID:Addgene_17361)	[38]

**Table 2 cells-09-02547-t002:** E3 ubiquitin-protein ligases identified by shotgun LC-MS/MS analysis of proteins co-immunoprecipitated with HBc-HA expressed in HepG2-hNTCP cells.

E3 Ubiquitin-Protein Ligase	HBc # Peptides Exp1/Exp2/Exp3	HBc # PSMs Exp1/Exp2/Exp3	CTRL # Peptides Exp1/Exp2/Exp3	CTRL # PSMs Exp1/Exp2/Exp3
Baculoviral IAP repeat-containing protein 6 BIRC6 *	122/109/124	219/171/204	0/0/0	0/0/0
E3 ubiquitin-protein ligase UBR4 *	96/75/90	140/101/121	0/0/0	0/0/0
E3 ubiquitin-protein ligase HUWE1 *	78/67/84	125/90/117	0/0/0	0/0/0
E3 ubiquitin-protein ligase UBR5 *	43/30/38	59/40/52	0/0/0	0/0/0
E3 ubiquitin-protein ligase TRIM21	26/24/26	57/51/47	26/24/21	62/49/50
E3 ubiquitin-protein ligase HECTD1 *	24/7/14	25/7/15	0/0/0	0/0/0
E3 ubiquitin-protein ligase UBR2 *	17/6/12	19/8/12	0/0/0	0/0/0
E3 ubiquitin-protein ligase TRIM71	7/2/4	9/2/4	1/0/0	1/0/0
Ubiquitin-protein ligase E3A	6/3/4	8/3/4	0/0/0	0/0/0
E3 ubiquitin-protein ligase MARCH7	5/2/0	5/2/0	0/0/0	0/0/0
E3 ubiquitin-protein ligase TRIP12	5/3/4	6/3/5	0/0/0	0/0/0
NEDD4-like E3 ubiquitin-protein ligase WWP1	5/1/4	5/1/4	9/3/7	10/3/7
E3 ubiquitin-protein ligase KCMF1 *	4/5/7	6/6/9	0/0/0	0/0/0
E3 ubiquitin-protein ligase RNF31 *	4/4/2	4/4/2	1/0/0	1/0/0
E3 ubiquitin-protein ligase Mdm2 *	3/3/4	3/3/4	0/0/0	0/0/0
E3 ubiquitin-protein ligase UBR1 *	3/2/2	4/2/2	0/0/0	0/0/0
Putative E3 ubiquitin-protein ligase SH3RF2	2/2/2	4/4/3	0/0/0	0/0/0
E3 ubiquitin-protein ligase RNF138	2/1/2	2/1/2	2/1/0	2/1/0
E3 ubiquitin-protein ligase RNF181 *	2/1/0	2/1/0	0/0/0	0/0/0
Probable E3 ubiquitin-protein ligase HERC1	2/0/0	3/0/0	0/0/0	0/0/0
E3 ubiquitin-protein ligase MIB1	2/0/1	2/0/1	3/0/3	4/0/4
NEDD4-like E3 ubiquitin-protein ligase WWP2	1/1/2	1/1/2	1/1/1	1/1/1
E3 ubiquitin-protein ligase AMFR	1/1/1	1/1/1	0/0/0	0/0/0
E3 ubiquitin-protein ligase RING1	1/2/2	1/2/2	1/2/0	1/2/0
E3 ubiquitin-protein ligase RBBP6	1/2/3	1/2/3	3/1/2	5/1/3
E3 ubiquitin-protein ligase Praja-1	1/0/1	1/0/1	0/0/0	0/0/0
E3 ubiquitin-protein ligase rififylin RFFL	1/0/0	1/0/0	0/0/0	0/0/0
E3 ubiquitin-protein ligase MYCBP2	0/0/3	0/0/3	0/0/0	0/0/0

HBc, HBc-HA sample; CTRL, negative control (empty vector) sample; # peptides, the number of detected peptide sequences unique to a protein; # PSM, the total number of identified peptide spectra matched for the protein. All analyses were performed in triplicate. * denotes E3 ubiquitin-protein ligases with 1.5-fold higher expression in HBc-HA sample with statistical significance *p* < 0.01 (Volcano plot Appendix A).

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
