# Peer review of "Hepatitis B Core Protein Is Post-Translationally Modified through K29-Linked Ubiquitination"

_cells, 2020, doi:10.3390/cells9122547_

Round 1

Reviewer 1 Report

In this work, Langerova et al. investigated posttranslational modifications (PTMs) of the hepatitis B virus core protein (HBc). PMTs are known to regulate numerous cellular processes that are captured by viruses for benefit. Inspired by their previous work and report (Ref. #31), the authors focused on the modification of HBc by ubiquitin and ubiquitin-like proteins. By using elegant and sensitive techniques, the authors succeeded to identify a K29 ubiquitin linkage, a quite atypical ubiquitination, attached to the lysine residue K7 of HBc. Thus far, a virus exploitation of K29 ubiquitin linkage has been merely described for the influenza A virus. Accordingly, the findings of this manuscript are very novel and innovative. In addition, by using MS analyses, the authors discovered distinct E3 ubiquitin ligases that may contribute to the K29-linked ubiquitination of the viral core protein. Despite my enthusiasm, the manuscript suffers from some experimental weakness that need to be considered and revised.

Major comments:

  1. HBc and its posttranslational modifiers (Chapter 3.1, Fig. 1). This chapter is a bit confusing and does not justify the conclusions drawn by the authors. By using CO-IP analyses, the authors aimed to examine, whether the UBL-proteins SUMO-1, SUMO-2, SUMO-3 and ISG15 could interact with HBc or covalently modify HBc. Thereby, “no interaction” could be detected (Figure 1). However, given that the cells were lysed under stringent conditions of denaturing buffer (2% SDS), “no interaction” is to be expectable. To detect an interaction, cells must be lysed under non-denaturing conditions. Moreover, the results shown in Figure 1 do not allow the conclusion that “HBc is not posttranslationally modified by the UBL-proteins”. The used tagged UBL-proteins have molecular weights of about 15 – 20 kDa (see Figure 1). If HBc would be modified, its molecular weight will shift to 35 – 40 kDa. This is not shown in the figure. Why are the blots (WB: anti-HBc ) are cut down and not totally presented? Overall, this chapter/issue requires an extensive revision.

  1. The use of epitope-tagged HBc constructs throughout the work. (i) The HBV capsid/nucleocapsid is built up by the single HBc protein. There is ample evidence that HBc is very sensitive towards mutations including insertions. Accordingly, I strongly recommend to prove that the C-terminally HA- and FLAG-tagged HBc constructs are still functional, i.e. competent for capsid assembly. This can be done by i.e. by performing capsid assembly assays. (ii) In addition, the authors should indicate the foreign amino acid sequences added to HBc. With regard to the FLAG-tag, used in all ubiquitination assays, there are serious concerns. What type of FLAG-tag was used, the canonical DYKDDDDK or tandem DYKDHD-G-DYKDHD-I-DYKDDDDK tags? Because both tags contain lysine (K) residues, they could serve as potential ubiquitination acceptor sites, thereby falsifying the experimental outcome. To univocally prove that HBc is a substrate for ubiquitination, at least one UB assay (e.g. Figure 3) must be performed in addition with untagged HBc (or alternatively tagged HBc without lysine residues). (iii) In this regard, the phenotype of the HBc-K7R mutant may be also misinterpreted. How do the authors exclude the possibility that the K7R mutation alters the conformation of HBc thereby rendering the lysine residues in the attached FLAG-tag less accessible for UB attachment?

  1. The UB pattern of the HBc-K7R mutant is not uniformly described in the text. Lane 313: HBc-K7R substantially reduced UBn; lane 334: HBc-K7R significantly increased UBn. In this respect (HBc-K7R), the data shown in Figures 3 and 4 are not consistent.

  1. Ubiquitination of endogenous HBc (Chapter 3.3, Figure 6). This is an very important experiment, but needs improvement. The IP/WB blot shows some UB-specific smearing signals. To ascertain that these signals are truly related to HBc-specific immunoprecipitation, the precipitation of HBc should be controlled by HBc-specific WB. Do the authors have quantification data? The conclusion that “replacing Ub-WT with Ub-K29R led to a reduction of HBc polyubiquitination” is not noticeable in the blot. 

  1. Table S2 may be shifted to the main manuscript (rather than Supplemental), as the results of this MS analysis are summarized in the abstract.

  1. To improve the discussion, the authors may also consider/discuss more clearly that the lysine K7 residue of HBc is not essential for HBV production and release. Hence, the K7-linked ubiquitination is unlikely to serve as an assembly advice. What other roles are conceivable?

Reviewer 2 Report

In this study Ludyova et al. investigate the role of uquitinylation in HBc post-translational modifications. As previously documented they show that HBc is poly-ubiquitinylated in transiently transfected-hepatoma cells. Using this same cell model, they further confirm the role or HBc lysine 7 in this phenomenon and also show that the formation of HBc-linked poly-ubiquitin chain occurs through lysine 29. Finally, they perform a preliminary mass spectrometry analysis in Huh7 and 293 cells to tentatively identify the ubiquitin ligase involved. All together this study does no answer a major question regarding the occurrence of this phenomenon during a bona fide viral infection and, more importantly, of its role. As such the title of the study is misleading.

Major comments

- Fig 2 and 3 show that HBc is ubiquitinated and that K7 is the major lysine involved in this PTM. These data are redundant with those previously published by the same authors in 2017 (Lubyova et al. PLoS One 2017, figure 7A and B) except that here transfection is performed in Huh7 cells instead of HepG2NTCP and that a quantification was performed to analyze the effect of these mutations on each single HBc-ubiquitin form.

- Fig5: the authors state that similar levels of ubiquitination are observed with Ub-wt and Ub-K29 but with the latter, the visualization of HBc signals (in red) comigrating with the Ub ones (in green) suggests that even higher levels of ubiquitination occur. Similarly, additional HBc bands (in red) migrating above the expected size are seen with several mutants (see for examples images with Ub-K33, -K48, and K-63). Are these bands ubiquitinated-HBc and if it is the case why the HBc-K7 mutant does not have any effect?

- Figure 6 shows that HBc can be ubiquitinated in infected cells when uquitin is over-expressed. What is the effect of this over-expression on viral replication? Does it modify HBc intra-cellular localization? Does it modify the other viral parameters?

- Why performing the mass spectrometry (Table S2) analysis in 293 cells? What is the relevance of this cell model? Are any of the E3-ubiquitin ligases identified using HBc as a bait, expressed in differentiated hepatocytes?To be informative such analysis should be performed in triplicate in Huh7 or HepG2 cells using a wt and a mutated (HBc-K7R) Core.

Minor points

Fig.1(a) and (d) : WB anti Myc instead of anti-FLAG?

Fig.6 there is evidently an inversion between lanes K29 and K29R

Reviewer 3 Report

In this manuscript, Langerova et al address the posttranslational modification of the HBV core protein by ubiquitination. The process occurs predominantly at the first lysine (K7) of core protein and is potentially mediated by UBR5 E3 ubiquitin ligase, through K29-linked ubiquitin. The methods are well described, in details, but there are some flaws as stated below:

Major comments:

1) In Fig 2, in the presence of MG132 inhibitor, no accumulation of core protein is observed, even when the Ub-WT is overexpressed. How can this be explained knowing that HBc is proteasomal degraded by NIRF?

2) The ubiquitination of HBc occurs (is detectable) only when the ubiquitin is overexpressed. However, the physiological importance of K7 ubiquitination in HBc protein could be demonstrated by infecting HepG2-NTCP cells with a virus containing K7R mutation in HBc and comparing the replication, assembly, or egress with the wild type virus. And why does HBc have a higher level of expression when co-expressed with ubiquitin, is it just a loading problem?

3) The data found in MS experiments should be validated by silencing or overexpressing the E3 ubiquitin ligases (UBR5, ITCH) found to interact with HBc, otherwise, the message is weak.

Minor comments:

1) Section 2.3 (Plasmids) is difficult to read due to the web page links. Please organize them in a table and keep only the Addgene numbers of the plasmids in the text.

2) Fig 3 and 4 state the same thing, i.e. fig 4 is a triplicate experiment of what is described in fig 3. I think they could be merged in one figure.

Round 2

Reviewer 2 Report

The authors answered to most of my questions. The only poin that, in my opinion, was not properly adressed concerns the effect of HBc uquitination on the viral life cycle.

Reviewer 3 Report

No further comments.